# Advances in NURR1-Regulated Neuroinflammation Associated with Parkinson’s Disease

**DOI:** 10.3390/ijms232416184

**Published:** 2022-12-19

**Authors:** Murad Al-Nusaif, Yushan Lin, Tianbai Li, Cheng Cheng, Weidong Le

**Affiliations:** 1Liaoning Provincial Key Laboratories for Research on the Pathogenic Mechanism of Neurological Diseases, First Affiliated Hospital of Dalian Medical University, Dalian 116021, China; 2Institutes of Neurology, Sichuan Academy of Medical Sciences & Sichuan Provincial Hospital, Chengdu 610072, China

**Keywords:** Parkinson’s disease, neuroinflammation, nuclear receptor-related transcription factor 1, microglia, astrocytes

## Abstract

Neuroinflammation plays a crucial role in the progression of neurodegenerative disorders, particularly Parkinson’s disease (PD). Glial cell activation and subsequent adaptive immune involvement are neuroinflammatory features in familial and idiopathic PD, resulting in the death of dopaminergic neuron cells. An oxidative stress response, inflammatory mediator production, and immune cell recruitment and activation are all hallmarks of this activation, leading to chronic neuroinflammation and progressive neurodegeneration. Several studies in PD patients’ cerebrospinal fluid and peripheral blood revealed alterations in inflammatory markers and immune cell populations that may lead to or exacerbate neuroinflammation and perpetuate the neurodegenerative process. Most of the genes causing PD are also expressed in astrocytes and microglia, converting their neuroprotective role into a pathogenic one and contributing to disease onset and progression. Nuclear receptor-related transcription factor 1 (NURR1) regulates gene expression linked to dopaminergic neuron genesis and functional maintenance. In addition to playing a key role in developing and maintaining neurotransmitter phenotypes in dopaminergic neurons, NURR1 agonists have been shown to reverse behavioral and histological abnormalities in animal PD models. NURR1 protects dopaminergic neurons from inflammation-induced degeneration, specifically attenuating neuronal death by suppressing the expression of inflammatory genes in microglia and astrocytes. This narrative review highlights the inflammatory changes in PD and the advances in NURR1-regulated neuroinflammation associated with PD. Further, we present new evidence that targeting this inflammation with a variety of potential NURR1 target therapy medications can effectively slow the progression of chronic neuroinflammation-induced PD.

## 1. Background

Parkinson’s disease (PD) is the most common form of movement disorder, and it is the second most prevalent neurodegenerative disease after Alzheimer’s disease [1], with a prevalence of about 0.5–1% among people aged 60–69 and rising to 1–3% over the age of 80 [2,3]. As the world’s population gets older, it is projected that the prevalence of PD will grow rapidly, doubling over the next two decades, resulting in direct and indirect impacts on the community and the economy overall [4]. The movement presentation is caused mainly by the degeneration of dopaminergic neurons in substantia nigra compacta (SNc), with intracellular aggregation of α-synuclein in Lewy bodies and Lewy neurites. Although neurodegeneration is not restricted to nigral dopaminergic neurons, it also affects cells in other areas of the neural network [5,6]. The interaction of genetic and environmental variables is still the most likely etiology of PD. Nevertheless, oxidative stress (OS), mitochondrial dysfunction, cellular calcium imbalance, neuroinflammation, neurotransmitter, and transcriptional factors (TFs) system dysregulation all contribute to the pathogenesis of PD [1,7,8]. PD is typically diagnosed based on the patient’s clinical history. Once motor symptoms occur, ~60 percent of dopaminergic neurons in the SNc have been lost [9]. Furthermore, patients frequently seek a diagnosis some years after the onset of motor symptoms. In clinical practice, diagnosing PD before motor symptoms is difficult due to a lack of distinguishing clinical and laboratory markers to detect pre-motor PD. This delayed diagnosis limits the development of disease-modifying therapies for reversing or slowing disease progression. Therefore, early detection of PD is essential and, if successful, will substantially benefit defining the therapy window and may lead to a better clinical outcome [10]. Likewise, new techniques like cell therapy and gene therapy are not used as much as they could be due to ethical and safety concerns. The development of innovative disease-modifying therapies for the treatment of PD is contingent on a better understanding of the disease’s pathogenic mechanisms. It is worth noting that both epidemiological and genetic research shows that neuroinflammation plays a critical role in PD pathogenesis and progression.

The central nervous system (CNS), which is insulated from the periphery immunological response via its blood-brain barrier (BBB), had previously been considered completely impervious. Conversely, this concept has been adjusted since the CNS target what is called the damage-associated molecular patterns (DAMPs) and pathogen-associated molecular patterns (PAMPs) by a robust nonspecific immune response [11,12]. Inflammatory pathways are involved in the dopaminergic neuron degeneration of PD and are highly influenced by genetic-environmental predispositions and brain immunological-induced glial activation [13]. In addition, active peripheral inflammation in PD exacerbates and synergizes with central inflammation to promote dopaminergic neurodegeneration [14,15]. Both post-mortem analyses of PD patients and animal experiments [16,17,18] showed that persistent activation of glial cells and the subsequent increase of pro-inflammatory substances exacerbate the SNc dopaminergic neuron loss. Additionally, microglial and astrocyte cells expressed the most PD-causing genes, which altered the protective role of these cells into a reactive and chronic one. TFs are up-regulated in the brain’s chronic neuroinflammation and enhance inflammatory cell activation, resulting in PD via autophagy of dopaminergic neurons and numerous other poorly known pathways [19]. Nuclear receptor-related transcription factor 1 (NURR1) is expressed throughout adulthood and plays a critical role in early development via a transcriptional mechanism to maintain the differentiation and maturation of dopaminergic neurons. Furthermore, NURR1 plays a critical role in the neuroinflammation and cellular metabolism of PD. In our narrative review, we conducted “a comprehensive PubMed search to find studies assessed the role of inflammation in PD and potential NURR1 involvement that were published in English using the title and abstract MeSH search (((Parkinson’s disease) OR (PD)) AND ((NURR1) OR (NR2A2))) AND (Neuroinflammation)/((Parkinson disease) OR (PD)) AND (Neuroinflammation), after that, a full-text read of the closely related articles. We will summarize the key cellular and molecular mechanisms driving neuroinflammation in PD, highlighting the role of central and peripheral inflammation in PD pathogenesis, and then provide a detailed discussion of the pathophysiological pathways linked between NURR1 and neuroinflammation. Finally, we will go through the role of NURR1 in potential target and disease-modifying therapeutics for overcoming or preventing inflammation in PD.

## 2. Inflammation and Immune Dysfunction in PD: A Cause or a Consequence?

The brain is increasingly recognized as a highly immunological-specialized organ with CNS-resident immune cells. Numerous studies have established the complex neuroimmune interactions in the CNS under both homeostatic and pathological conditions, and the concept of the CNS as an immune privilege has been refined [20]. Excessive neural inflammation is a major contributing factor in neurodegenerative disorders; in PD, the BBB integrity is compromised, which may be preceded by the activation of the innate immune system, permitting the enrollment and stimulation of the adaptive immune mechanism [20,21]. Deregulation of central and peripheral inflammatory pathways is one of the components of PD pathogenesis [22], which most likely arises due to genetic predispositions in conjunction with immunological changes associated with age and the main activation of glia of neuronal injury (Figure 1).

### 2.1. Microglial Implication in PD

Microglia demonstrate a homeostatic function in basal conditions, assess the microenvironment insults, and respond to inflammatory molecules [23]. Microglia is conventionally characterized by two distinct phenotypes: classical pro-inflammatory and anti-inflammatory phenotypes [24]. Along with these modifications, microglia increase the expression of inflammatory cell surface markers such as major histocompatibility complex I/II (MHC I/II), chemokines, and cytokines [25]. McGeer et al. found reactive microglia positive for human leukocyte antigens D-related isotype (HLA-DR+) in SNc of post-mortem PD patients [26]. Throughout the whole nigrostriatal pathway, HLA-DR+ reactive microglia are more abundant in association with neuronal degeneration [27]. DAMPs released by dying neurons (ATP, neuromelanin, m-calpain, matrix metalloprotease 3) or pro-inflammatory mediators secreted by astrocytes chemokine (C-C motif) ligand 2 [28], misfolded or aggregated proteins, such as α-synuclein, along with signals delivered by Toll-like receptor (TLR) [25,29], favor the acquisition of the pro-inflammatory microglia. The pro-inflammatory microglia tended to result in the upregulation of MHC I and II complex [26]. They increased pro-inflammatory mediator cytokines, such as (interleukin-1 beta (IL-1β) and interleukin-6 (IL-6), tumor necrosis factor-alpha (TNF-α), chemokines, and bioactive lipids) [30]. Therefore, pro-inflammatory microglia could change the BBB’s permeability and cause circulating leukocytes to infiltrate the brain [31], amplifying the local inflammatory response.

Previous research demonstrated that inwardly rectifying potassium (Kir6.1) channels expressed in microglia and the opening of the microglial adenosine triphosphate (ATP)-sensitive potassium channel might decrease neuroinflammation and mitigate rotenone-induced degeneration of dopaminergic neurons [32]. In a mouse PD SNc model, Kir6.1 promoted anti-inflammatory microglia polarization, and Kir6.1 deficiency increased dopaminergic neuron death and microglial activation. Conversely, inhibiting p38 mitogen-activated protein kinase (MAPK) alleviated the negative effects of Kir6.1 deletion on microglia and dopaminergic neurons [33]. Additionally, a recent study showed that Kir6.1 is a negative regulator of the nucleotide-binding oligomerization domain-like receptor protein 3 (NLRP3) inflammasome [34]. This inhibition of inflammasomes is presumably mediated by the interaction between Kir6.1 and NLRP3. These data also show that Kir6.1 is a prospective target for treating inflammasome-mediated inflammatory illnesses such as PD. Furthermore, microglia exposed to α-synuclein create a cellular network by forming F-actin-dependent intercellular connections that transmit α-synuclein from overloaded microglia to adjacent homeostatic microglia [35]. Reducing the α-synuclein load decreased the inflammatory profile of microglia and increased their survival [35]. To put it simply, persistently reactive microglia produce vast quantities of pro-inflammatory mediators that damage neurons and stimulate further microglia activation, causing a vicious cycle of neurodegeneration and neuroinflammation.

### 2.2. PD-Related Astrocytic Inflammation

Despite receiving far less attention than microglia in the context of PD, astrocytes are now more commonly thought to be implicated in disease progression. Under pathological conditions and in response to inflammatory processes, astrocytes may interact with microglia to promote the immune response and activate apoptotic pathways [36], resulting in the death of dopaminergic neurons. Reactive astrocytes generated in the CNS in response to stimuli or lesions contribute to the etiopathogenesis of PD via toxic gain-of-function [37]. These reactive astrocytes release chemokines and cytokines, including TNF-α and IL-1β [38,39]. Lipopolysaccharide (LPS) is a potent glial activator; in vitro treatment of animals lacking microglia or pure astrocytes fails to be stimulated. The Liddelow group reveals that LPS activates astrocytes by prompting microglia to release IL-1α, TNF-α, and complement component 1, subunit q [40], thereby shifting astrocytes into a reactive state.

A noteworthy finding was the presence of reactive, neurotoxic astrocytes in post-mortem PD brains [40] and animal models of PD [41]. When the PD-related A53T mutant α-synuclein is selectively expressed in astrocytes [42], mice acquire rapidly progressive paralysis, astrogliosis, microglial activation, and dopaminergic neuron degeneration. These findings show that defective astrocytes and reactive astrogliosis play a role in the development and progression of PD. We study the regional susceptibility to aging and age-related disorders. A recent review [43] summarized regional astrocyte variability across brain areas, revealing cellular, molecular, and functional heterogeneity. These localized astrocyte variations may contribute to selective susceptibility in PD, and the link between those changes and aging or disease progression should be explored further. It has been proven that microglia and astrocytes play important roles in CNS homeostasis and that these neuroprotective roles are compromised when the brain is injured. Furthermore, both glial cells interact with activated microglia leading astrocytes to have a neurotoxic phenotype [40]. Glia activation prevention positively affects neuronal survival when many agonists and TFs, including NURR1, are used, as will be discussed further below.

### 2.3. Endothelial Inflammation Impact on PD

The brain’s endothelial cells serve as a core component of the neuronal and vascular unit and strive to maintain homeostasis in the CNS. Despite having a protective function, endothelial cells also regulate leucocyte adhesion, vascular permeability, thermogenesis, and the nutrition supply. Breaking the endothelial barrier and migrating inflammatory cells across the endothelial layer is critical for CNS inflammation [44]. During the migration, a complex process of adhesion compounds and interaction mechanisms occurs at the junction of the endothelium to accomplish the cross-to-CNS barrier and then neuroinflammation [44]. BBB endothelial cells react to peripheral encounters. When rhesus monkey isolated endothelium cells were exposed to LPS or hypoxia, the results showed an increase in the release of IL-6. The older endothelial cells responded more robustly by releasing IL-6 [45]. Brain endothelial cells, like glial cells, express TLR [46] and respond to DAMPs and PAMPs with an endothelium-derived inflammatory response. The release of adhesive molecules, chemokines, and pro-inflammatory cytokines from the injured endothelium is thought to cause endothelial dysfunction [44].

Vasculature alterations in PD are connected with striatal and midbrain BBB leakage. PD patients had a positron emission tomography scan, which showed a Cluster of differentiation 4/8 (CD4+ and CD8+) infiltration linked with increased BBB permeability and cell loss [47]. In PD, the SNc, white matter, and posterior cortical regions exhibited subtle BBB disruption [48]. Additionally, due to arteriolar fragmentation and the loss of capillary connections, the vasculature in the SNc of PD patients has a defective morphology characterized by endothelial “clusters [47]”. Other research indicated that brain endothelial cells actively modulate immune responses to CNS inflammation, as shown by the impairment of endothelial function by proinflammatory cytokines like IL-1β or TNF-α [49]. PD patients had significantly higher plasma levels of the soluble very late antigen 4 ligands soluble vascular cell adhesion molecule-1(sVCAM1) than age-matched healthy controls [50]. In addition, the VCAM1 was correlated with disease severity, quality of life, and non-motor symptoms. A disruption of the BBB is observed in PD animal models, which has recently been supported by the use of dynamic contrast-enhanced magnetic resonance to assess the integrity of the BBB in PD in comparison to age-matched cerebrovascular disease and healthy control. For a further understanding of endothelial inflammatory pathogenesis, further research is needed, mainly in the initiating the endothelial damage and lack of BBB and their link to the inflammasome. These include longitudinal clinical imaging studies combining neuronal, metabolic, and vascular parameters.

### 2.4. Peripheral Inflammation and PD

The signaling of activated innate and T cells in the enteric nervous, digestive, and circulatory systems is called “peripheral inflammation” [51]. In many neurodegenerative disorders, peripheral inflammation aggravates underlying central degeneration [13,52], resulting in multiple reactions and signaling, including proinflammatory substances entering the CNS from peripheral systems, T cells, mast cell infiltration, chemotaxis delivery or disruption of the BBB, and persistent glial cell activations. In addition, nitrated α-synuclein can circumvent or overcome immunological tolerance and activate peripheral leukocytes (initiating peripherally-driven CD4+ and CD8+ T cell responses) [53], hence accelerating the degeneration of nigral dopaminergic neurons. Astrocytes can be directly activated by circulating molecules and secrete pro-inflammatory molecules and neurofilaments. Microglia can be induced by neuronal MHC-I found in dopaminergic neurons, which is initiated by either α-synuclein, neuromelanin, or cytosolic dopamine (DA) overexpression and/or OS and subsequently targeted by cytotoxic T cells [54].

Brain-gut-microbiota axis in PD is another connection between the peripheral and CNS. The overstimulation of the innate immune response by gut dysbiosis and/or an increase in the intestinal microbiota combined with intestinal permeability disruption may result in systemic inflammation [55]. In contrast, enteric glial and neuronal cell activation may promote the initiation of alpha-synuclein aggregation. As a result, detecting α-synuclein alterations in the gut is critical for determining its involvement in the intestinal neurological circuit and its transmission from the gut to the nervous system, which has heightened interest in the early detection and diagnosis of PD. Moreover, bacterial proteins cross-react with human antigens and can disrupt the adaptive immune system [55,56]. In summary, central and peripheral cellular and molecular inflammatory changes are imperatively recognized as risk factors in the pathogenesis and progression of PD (Table 1). A deeper insight into the strategies implicated in glia activity and their conversion to an inflammatory phenotype should lead to a better understanding of the pathology of the disease and future therapeutic options.

## 3. Nuclear Receptor Related-1 and Neuroinflammation Associated with Parkinson’s Disease

As previously explained, numerous studies show evidence that neuroinflammation is relevant to dopaminergic neurodegeneration and PD progression. Numerous studies postulate that the etiopathogenesis of PD may be influenced by the interaction of genetic and neuroinflammatory factors [90,91,92]. NUUR1 is a member of the nuclear receptor subfamily 4A (NR4A), which includes NURR1, NUR77, and NOR1 (also known as NR4A2, NR4A1, and NR4A3, respectively). In neurodegenerative disorders, these NR4A proteins are involved in neuroinflammation and cell death [93]. NURR1 is expressed early in embryogenesis and is required to maintain maturing and adult midbrain dopaminergic neurons [94,95]. It is widely expressed in the CNS, mainly in the SNc, ventral tegmental area (VTA), and limbic region [96,97]. NURR1 is also expressed in microglia, peripheral blood mononuclear cells (PBMCs), endothelial cells, and other non-neuronal cells [96,98]. The broad expression of NURR1 influences the course of numerous CNS and organ tissue disorders. It plays a role in the pathogenesis of CNS diseases such as neuroinflammation [99], PD [81], AD [100], multiple sclerosis [101], schizophrenia, and manic-depressive disorder [102]. It also plays a role in rheumatoid arthritis [103] by modulating pro-inflammatory genes and synovial inflammatory processes. An ever-expanding body of research has firmly proven the importance of NURR1 in the pathogenesis of neurodegenerative diseases and the development of innovative treatment strategies. We explore the structural characteristics of NURR1 that are critical for its function in neuroinflammation and dopaminergic neuron death in PD. In addition, we highlighted the neuroprotective and anti-inflammatory components of NURR1 and their therapeutic potential for PD.

### 3.1. Functions of Nuclear Receptor Related-1 in Dopaminergic Neurons and PD

NURR1 is essential for developing and maintaining midbrain dopaminergic neurons [104]. We first reported that two human mutations in exon 1 of the *NURR1* gene resulted in decreased *NURR1* mRNA expression in familial PD [105]. Then our team group and others found that dopaminergic neuronal expression of *NURR1* declines with normal aging [106] and further decreases in the PD patient’s SNc and peripheral blood [107,108]. The expression of functional DA genes, such as tyrosine hydroxylase (TH), vesicular monoamine transporter 2, aromatic amino acid, and DA transporter, is regulated by the NURR1 [96] (Figure 2). Homozygous mice knockout (KO) for *Nurr1* in the germ line fail to develop dopaminergic neurons, and they die soon after birth [109,110]. *Nurr1* heterozygous mice, on the contrary, survived with locomotor deficits linked with decreased striatum DA and high vulnerability to neurotoxic assaults [111]. Furthermore, conditional knockout (cKO), the nigrostriatal pathway of *Nurr1* mice, selectively ablates the dopaminergic neuron *Nurr1,* reflecting similar biochemical and pathological characteristics of PD [94,112]. *Nurr1* cKO mice mimic the early aspects of PD, demonstrating locomotor deficits and a series of pathological alterations such as loss of dopaminergic neurons and decreased striatal DA, which might serve as a valuable model to investigate PD [112].

NURR1 transcriptionally controls several target genes in dopaminergic neurons, including kelch-like 1, guanosine-5-triphosphate, delta-like non-conical notch ligand 1, vasoactive intestinal peptide, and protein tyrosine phosphatase receptor type U [113,114,115]. The paired-like homeodomain 3 (PITX3)/Wingless-related integration site/β (Wnt/β)-catenin transcriptional pathways are two key signals that cooperate with NURR1 with the NURR1 transcription complex to promote midbrain dopaminergic neurogenesis [109,116]. NURR1 and PITX3 allegedly regulate genes involved in critical dopaminergic neuron processes such as neuronal patterning, axonal growth, and terminal differentiation [113]. Growing evidence indicates a complex network between NURR1 and other important transcriptional elements during dopaminergic neuron development and in neuronal pathologies such as experiencing PD. These findings shed light on the critical role of NURR1 in the dopaminergic neuron and its extensive involvement with other TFs, which pave the way for future research in this direction to understand the pathogenesis mechanism of PD.

### 3.2. Involvement of Nuclear Receptor-Related Factor 1 in PD Neuroprotection and Neuroinflammation

One of the causes of neurodegeneration is chronic inflammation. In PD, dead or injured dopaminergic neurons can directly activate microglia, producing a large number of reactive oxygen species (ROS) and pro-inflammatory cytokines. Several TFs and signaling pathways intermediate the pathological process of neuroinflammation and modulate the genetic and environmental causes of neurodegeneration diseases. NURR1 [99] is one of the TFs and signaling pathways involved in PD-related neuroinflammation mechanisms (Figure 2). Studies have revealed that NURR1 is expressed in activated microglia, astrocytes, and macrophages [98,117,118]. Dopaminergic neurons are protected from the inflammatory process by inhibiting the production of pro-inflammatory factors in these cells. Compared to LPS-injected mice alone, the SNc of adult mice injected with lentivirus-encoding *Nurr1* followed by LPS injection reduced TH neuron numbers while increasing microglial TNF-α and IL-1β levels [36]. Conditioned media from LPS-treated glial cultures, which contain IL-1 β and TNF-α, causes cultured dopaminergic neurons’ death [119]. Additionally, in adult rat SNc, the sustained IL-1β expression using a recombinant adenovirus leads to the death of dopaminergic neuronal cells [120].

LPS-treated astrocytic and microglial cell culture validated the NURR1’s anti-inflammatory function via interacting with inflammatory nuclear receptors through nuclear factor-kappa (NF-κB)-p65. The corepressor for element-1-silencing TF corepressor complex is then recruited by *Nurr1*, reverting the expression of NF-κB-activated genes to baseline [36]. Another shared pathogenic feature of neurodegenerative diseases is the abnormal accumulation of protein aggregates in neuronal and glial cells. While it has been stated that the misfolded protein can either directly or indirectly activate glial cells, it is worth noting that *Nurr1* inhibits α-synuclein transcription [121] and vice versa, which may affect disease progression. Our most recent research has shown that α-synuclein at the promoter region located at −605 and −418 bp, which contains NF-κB, modulates the transcription of *Nurr1* [122]. Moreover, the α-synuclein expression reduces the quantity of the binding site of NF-κB to the *Nurr1* promoter, leading to reduced transcription of *Nurr1* [122] and perhaps activating proteasome-dependent NURR1 degradation in midbrain dopaminergic neurons [123].

Neuronal and glial cells produce prostaglandin E (PGE), a critical component in regulating inflammatory responses. PGE1 and its dehydrated metabolite, prostaglandin A1 (PGA1), communicate with the ligand-binding domain of *Nurr1*, which stimulates the transcriptional function of *Nurr1* [124]. In the same direction, recent research using nuclear magnetic resonance titration and crystal structures showing physical binding with PGA2 stimulates *Nurr1* transcription [125]. Furthermore, in leucine-rich repeat kinase 2 G2019S transgenic fly models, PGA2 can reverse locomotor impairments and neuronal degeneration [125]. In addition, PGE2 activation of the prostanoid early expanded protein 1 (EP1) receptors lead to an increase in the *Nurr1* expression by sequential stimulation of Ras homologous, protein kinase A, cyclic adenosine monophosphate-response element binding protein (CREB), and NF-κB signaling pathways which involved in the regulation of various cellular processes as cell morphology, adhesion, proliferation, and survival [126]. These findings suggest that the up-regulation of *Nurr1* by the EP1 and PGE1 receptors would have neuroprotective effects on the brain. Nevertheless, previous research found that activation of EP1 receptors in the CNS is associated with neurotoxicity and pharmacological blockade of the EP1 receptor exhibits dopaminergic neuron protection [127,128]. Nevertheless, it is currently uncertain if the up-regulation of Nurr1 promotes or interferes with this neurotoxicity. It would be interesting to see what future research uncovers about this dilemma.

#### 3.2.1. The Role of NURR1 in α-Synuclein-Mediated Inflammation

In synucleinopathies disorders, the transference of α-synuclein from neurons to glial cells causes inflammatory responses. Changes in the gene expression patterns of astrocytes subjected to neuronal α-synuclein indicate the development of an inflammatory response [129]. This was shown to be the case by the astrocytes turned up the expression of cluster of differentiation 14, NF-κB-1/-2, toll-like receptor 2, and REL-associated protein involved in NF-κB heterodimer formation, also known as p65 [129]. NF-κB complex activation promotes the transcription of several cytokines and chemokines [130], stimulating key components of inflammation and innate immunity-related circuits. A recent study found that NURR1 and NF-κB were important participants in the Toll-like receptor 4 (TLR4) NF-κB signaling process. NURR1 reduced TNF-α production by interacting with NF-κB/p65 and blocking its translocation [131]. Inhibiting the TLR4/NF-κB pathway and overexpressing *Nurr1* can reduce the inflammatory response to α-synuclein. The TLR4/NF-κB/NURR1 signal pathway involves the α-synuclein-induced microglial activation [131]. The model provides a valuable prediction for a new NURR1 inflammatory pathway where medications interact with the TLR4/phosphoinositide 3-kinase (PI3K)/serine/threonine protein kinase (AKT)/glycogen synthase kinase-3β signal pathway may affect neuroinflammation in α-synuclein-related neurodegenerative illnesses.

α-Synuclein overexpression and accumulation may disrupt NURR1 function by interfering with the signaling of a glial-derived neurotrophic factor (GDNF) [123,132]. The capacity of GDNF to have neuroprotective effects in PD is compromised by its interaction with α-synuclein [133]. In the neurons subjected to excitotoxic and OS, NURR1 acts as a mediator of CREB protein-dependent neuroprotective responses and a regulator of genes involved in neuroprotection [134]. Moreover, NURR1 also induces brain-derived neurotrophic factor (BDNF) production [135]. N-methyl-D-aspartate (NMDA) receptor activation enhances neuronal survival during brain development, and NURR1 has been linked to the activity-dependent survival of glutamatergic neurons. The pro-survival properties of NMDA on neurons require a boost in BDNF, and this enhancement is mediated by NURR1, which has been recognized as a ligand for CREP [136]. Furthermore, NURR1 regulates the GDNF receptor’s transcription and the rearranging during transfection (Ret)-protein’s C-terminal portion [137,138]. Ret tyrosine kinase is a high-affinity GDNF receptor complex anchored to the cell membrane by glycosylphosphatidylinositol [137]. GDNF protects mesencephalic dopaminergic neurons against neurotoxic insult, cell death, and apoptosis. The autophosphorylation following the GDNF binding the Ret tyrosine domine activates diverse neuron growth and survival systems, particularly the MAPK and PI3K [95]. Even though it has been demonstrated that BDNF produced from glial cells is effective in rodents and nonhuman primates, therapeutic findings in human studies have been inconsistent. It is hypnotized in the new systemic review that the failure of GDNF in clinical trials succumbed to the NURR1 shortfall, and future studies may help to cut the end. Despite these pathways of NURR1’s significantly contributing to abnormal α-synuclein aggregation and chronic glial activation, the precise association mechanism between disease-associated protein aggregation and neuroinflammation mediated by glial cells remains unknown.

Apart from the aforementioned, many others conducted extensive research to investigate the modulation mechanism of NURR1 in neuroinflammation. In primary microglial cells, we found *Nurr1* expression is enhanced after LPS treatment, and NURR1 translocated from cytoplasm to nucleus, which may be regulated by the PI3K/protein kinase B/Akt and the MAPK pathways [98]. Phosphorylation of extracellular signal-regulated kinase (ERK) can also increase *Nurr1* expression [98,139,140], which is further supported by a recent study that found a correlation between the phosphorylation of ERK and *Nurr1* [141]. A potential mechanism for the role of NURR1 in inflammatory settings induced by LPS showed that NURR1 could negatively control Ras guanyl-releasing protein 1 (RasGRP1) expression and exert its anti-inflammatory by inhibiting RasGRP1 expression; thereupon, the expression of inflammatory cytokines and the associated signaling cascade were reduced [142]. Multiple lines of evidence support NURR1 role in regulating immune cell function and modulating inflammation [143,144]. According to the Yoshimura and Sekiya groups [145], NR4A receptors play critical roles in controlling the fates of CD4+ T cells in the thymus and thus contribute to immunological homeostasis. NURR1 uniquely binds to the Foxp3 promoter directly, causing transcription activation and Regulatory T (Treg) cell formation. In addition, in T cells deficient *Nurr1*, abnormal Th1 induction is increased, while Treg cell induction is reduced [143]. NURR1 not only regulates the formation and suppression abilities of Treg cells, it likewise plays a vital role in inhibiting the induction of abnormal Th1. Moreover, NURR1 can regulate Th17 cell-mediated autoimmune inflammation [146] and prevent neuronal death by suppressing the activation of inflammatory genes in microglia and astrocytes [36].

#### 3.2.2. The Impact of NURR1 in Neuroinflammation Caused by Mitochondrial Dysfunction and Oxidative Stress

The intricate interplay between mitochondrial disarrangements and OS has been identified as a neurodegenerative contributor [147]. OS mainly causes damage to mitochondrial DNA (mtDNA), and the accumulation of the mtDNA is associated with aging and aging-related disorders [148]. The mtDNA can activate the TLR pathway through endolysosomal toll-like receptor 9 (TLR9) binding. Correspondingly, TLR9 recruits innate immunological signal transduction, which activates the MAPK and causes inflammation and neutrophil recruitment via NF-κB signaling [149,150,151,152]. Furthermore, mtDNA activates NLRP3 inflammasome, leading to the generation of interleukin-18 (IL-18) and IL-1β [153]. In addition to cytokine production, the activation of NLRP3 increases lysis and activation of pro-cysteinyl aspartate proteinases-1 (Caspase-1), ultimately promoting inflammation. When reactive microglia release pro-inflammatory cytokines, a cascade of signaling molecules are activated, resulting in the persistent release of ROS and cytokines, which cause neurodegeneration [154]. It is conceivable that neuroinflammation could impact mitochondria and change their function, given that such compounds have a high potential to hinder and injure mitochondria.

NURR1’s mitochondrial protective mechanism is accomplished by regulating mitochondrial-encoded nuclear genes, as with cyclooxygenase-β, mitochondrial translation elongation factor, and superoxide dismutase [107,155]. It has already been proposed that decreased *Nurr1* activity is linked to mitochondrial function instability [156], which accelerates dopaminergic neuron death [157]. Studies have demonstrated that deletion of NURR1 from dopaminergic neurons causes over 90% of the respiratory chain genes to be down-regulated [112]. Given that NURR1 is crucial for preserving dopaminergic neurons’ respiration processes, research has shown that deleting *Nurr1* in mice reproduces the initial manifestation of PD [112]. Concurrently, 1-methyl-4-phenylpyridinium (MPP+)-pretreated PC12 cells with peroxisome proliferator-activated receptor gamma and NURR1 agonists, either alone or in combination, caused a decrease in mitochondrial membrane potential and intracellular ROS, which in turn resulted in a reduced cell death rate [158]. The retinoic acid X receptor (RXR) ligand bexarotene, which forms a heterodimer with NURR1 and other TFs, co-regulates NURR1 target genes [159]. It has been shown that the synthetic agonist of NURR1/RXR, HX600, can inhibit the expression of microglia pro-inflammatory mediators and prevent cell death caused by inflammation [160]. Vassilatis et al. found that activating the NURR1/RXR-α heterodimer as monotherapy for PD may have a dual role by increasing striatal DA levels and up-regulating NURR1 target genes [161]. NURR1 may regulate inflammation via mitochondrial disarrangements and its gene pathway. More research is needed to comprehend the mechanism of NURR1 in this direction.

#### 3.2.3. Pyroptosis in PD and NURR1 Potential Impact

In chronic inflammation, the deregulation of pyroptosis produced by toxic signals such as microbial infection or aberrant protein accumulation is often associated with immunological dysfunction [162]. A hyperactive inflammatory reaction can result in a variety of neurological disorders. In recent years, numerous research findings have surfaced, indicating that activation of the inflammasome and pyroptosis are implicated in PD. For instance, aberrant accumulation of α-synuclein triggers activation of the NLRP3, which binds to caspase-1, enhancing the secretion of the inflammatory cytokines IL-1β and IL-18 and pyroptosis induction. Subsequently, the microglia were activated to release IL-1β [163]. Indeed, NLRP3 enhances mitochondrial permeability transfer pore opening, which facilitates the release of mtDNA [164]. Once mtDNA is released, it preferentially binds to NLRP3 and may serve as the ultimate NLRP3 ligand. This confirms ROS’s impact in activating the inflammasome in response to mtDNA [165]. Therefore, the mtDNA, ROS, and the steady cycle of mitochondrial dysfunction triggered by NLRP3 were assumed.

The function of NURR1 in pyroptosis has been the subject of significant progress in recent years. NURR1 downregulation enhanced the Müller cells’ activation by modulating the NF-κB/NLRP3 inflammasome axis [166]. Consequently, NURR1 might control an inflammatory response through the NF-κB/NLRP3 inflammasome. Itaconate, a powerful inflammatory modulator [167], in vivo and in vitro PD models can reduce neuroinflammation by suppressing the NLRP3 inflammasome [168]. Itaconate has neuroprotective effects in MPP+SH-SY5Y cells by restoring TH and NURR1 expression, both initially inhibited by MPP+ [168]. New research in a PD mouse model shows that the DA agonist pramipexole suppresses the astrocytic NLRP3 mediated by DA receptor D3 protein-dependent autophagy [169]. Research findings in PD-associated pyroptosis have primarily concentrated on the participation of the inflammasome. However, additional research is needed to determine the relevance of other, less-studied inflammasomes, particularly the part of the inflammasome and NURR1 present in various CNS neurons. Specifically dedicated to advancing cell-specific KO mouse approaches as well as single-cell RNA sequencing and mass cytometry.

#### 3.2.4. Alterations of NURR1 and Cytokines in PD

Cytokines and peripheral inflammatory cells are recognized as crucial factors in the neuroinflammatory cascades linked to the degenerative process in PD [170]. In the brain and cerebrospinal fluid of PD patients, numerous researchers have found elevated levels of inflammatory cytokines, i.e., TNF-α, IL-1β, interleukin-4, and IL-6 [171,172,173]. Furthermore, several studies have reported similar protein-level findings from serum samples or plasma [174,175,176]. However, the exact mechanism of elevated peripheral blood cytokine levels in PD patients is still unclear. It has been proposed that neuroinflammation in the CNS of PD patients may trigger systemic inflammatory responses and activate peripheral cells to express and produce more cytokines during the disease’s development and progression [85,177].

Our previous studies showed that the expression level of *NURR1* in PBMCs of PD patients was significantly lower than that of healthy controls and patients with other neurological diseases [81,108,178]. Furthermore, we recently demonstrated that the levels of *NURR1* in PBMCs from PD patients were inversely correlated with TNF-α, IL-1β, IL-10, and IL-6 [81]. This clinical change in *NURR1* and cytokines reinforces previous evidence that NURR1 plays a role in neuroinflammatory processes in PD patients. Interestingly, *NURR1* expression decreased with advanced disease onset and increased disease severity. *NURR1* expression, regardless of drug type, was significantly reduced in treated patients but not in untreated patients compared with healthy controls [81]. Consistent with these findings, Saijo et al. [36] demonstrated that NURR1 functions in microglia and astrocytes to inhibit the production of inflammatory mediators and protect against the degeneration of dopaminergic neurons. Another study concluded that NURR1 overexpression dramatically reduced the pro-inflammatory cytokines that reactive microglia produced [174]. Although statistical analysis demonstrated a negative link between NURR1 and cytokines, it is still unknown how NURR1 controls neuroinflammation in peripheral circulation, and further research will be required to elucidate this concern. Additionally, more research is needed to understand the underlying mechanisms of anti-parkinsonian drugs that may interact with NURR1 and cytokines.

## 4. NURR1 as a Potential Neuroprotective and Anti-Inflammatory Target Therapy

As it stands, most clinically-applied treatments for PD aim at alleviating symptoms. Unmet needs include disease-modifying therapies that can effectively alter the course of PD by delaying, halting, or reversing disease progression. However, there has not been notable success in disease modification. Developing novel treatment techniques based on innovative molecular mechanisms and targets is urgently required [179]. Multiple pathogenic factors have been identified in PD [180]. Consequently, therapies that can directly or indirectly modulate immune function are essential to developing therapeutic compounds for PD [181,182]. Besides its distinct physiological roles in dopaminergic neuronal differentiation and survival, NURR1 is vital for signal transduction pathways in mammalian organ systems. As per the mounting data, chronic innate neuroinflammation mediated by microglia and astrocytes is a cornerstone of PD and plays a complex role in its pathophysiology. The pathophysiology of PD is influenced by neuroinflammation and neuronal cell death. Accordingly, identifying medications with promising anti-inflammatory may be potential treatment options for PD. In this context, NURR1 inhibits and regulates various pro-inflammatory mediators in microglia and astrocytes; hence, NURR1 may be a viable neuroprotective target involving multifaceted processes [183,184].

A growing body of evidence that NURR1-activating agents, modulators, and *Nurr1* gene therapy might improve DA release and empower dopaminergic neurons against environmental toxins and reactive microglia [183,184]. In addition, the fact that these molecules modulate autophagy raises the prospect of a link between NURR1 and autophagy, which would explain why they are effective in treating PD [185]. The diverse functions of Nurr1 consider it an intriguing candidate for treating PD (Table 2).

Stem cell therapy for PD treatment has made great progress in the last three decades. Owing to their self-renewing and pluripotent properties, neural stem cells (NSCs) are potentially transplantable cells [186]. However, due to the inflammatory environment’s toxic effect, most transplanted NSCs transformed into glial cells rather than neurons, and only a few transplanted NSCs survived [187,188]. Therefore, addressing the inflammatory host brain environment improves the abnormal behaviors in PD rats. Therefore, it is of utmost importance to find a way to shield dopaminergic neurons from neuroinflammation and promote their survival. NURR1-co-grafting cell transplantation has fascinated attention as one of the most effective and appealing techniques. Several studies have demonstrated the therapeutic potential of NURR1-based cell transplantation in correcting behavioral abnormalities and modifying inhospitable host conditions [189]. Co-transplantation of Nurr1 overexpressed microglia with NSCs enhanced transplanted NSC survival and dopaminergic neuron growth by inactivating microglia and reducing inflammatory cytokines. Additionally, the transduction of NURR1 induces the development of dopaminergic neurons [190] (Table 2), which can survive and recover motor function in animal models of PD after transplantation. Combining NURR1 and other TFs with pluripotent stem cells or direct reprogramming of astrocytes or fibroblasts into human dopaminergic neurons has shown promising results. To conclude, it is proposed to explore more of the available agonists and modulators in preclinical and clinical research to identify a specific dose and time-dependent manner for activation of NURR1. The NURR1-targeting gene and cell-based therapy offer a promising prospect for next-generation PD clinical trials and should be investigated further due to safety, success rate, and ethical concerns.

**Table 2 ijms-23-16184-t002:** A summary of the potential neuroprotective and anti-inflammatory therapy roles of NURR1.

NURR1-Activating Compounds with Potential Neuroprotective and Anti-Inflammatory Target Therapy				
Compounds	Models	Methods	Main Outcomes	Ref.
AQ/CQ/HCQ	In vivo	AQ in 6-OHDA- lesioned rats	Interacts with LBD to modulate NURR1 transcriptional function and induces *Nurr1* to suppress pro-inflammatory cytokine gene expression in microglia	[191]
	In vitro	CQ in T cells	Promote *Nurr1*’s transcriptional activity by binding to LBD and up-regulating the expression of *Nurr1*, activating TREG cell differentiation.	[192]
	In vivo	HCQ in rat rotenone model	NURR1 expression is increased, NF-κB and pro-inflammatory cytokines (TNF-α, IL-1β) are inhibited, and GSK-3β activity is reduced.	[193]
1,1-bis(3′-indolyl)-1-(p-chlorophenyl) methane	In vitro	BV-2 reactive microglia (using LPS)	Enhance binding of NURR1 to the P65-binding site, reduce binding of P65 to inflammatory gene promoters, inhibit NF-κB-dependent gene expression	[194]
Daphnane-type and phorbol-type diterpenes	In vitro	BV-2 microglia cells	Activate *Nurr1* and inhibit LPS-induced nitric oxide production	[195]
Isoxazolo-pyridinone analog	In vivo	Administered in C57BL/6 lactacystin-lesioned mice	Increase expression of *Nurr1* and inhibit reactive microglia	[183,196]
SA00025	In vivo	Administered in 6-OHDA rats primed with the toll-like receptor 3 double-stranded RNA inflammatory stimulant	Decrease the reactive microglia and IL-6.	[138]
Cilostazol	In vivo	Orally administered daily in rotenone rats	Hamper the NF-κB and TNFα, IL-1β, up-regulate *Nurr1* and inhibit GSK-3β.	[197]
**Nurr1 Modulators and Neuroprotective and Anti-Inflammatory Role**				
**Compounds**	**Models**	**Methods**	**Main Outcomes**	**Ref.**
Memantine	In vitro	Administered in PC12 cells induced by 6-OHDA	Up-regulate NURR1 and eliminate the IL-6 and TNF-α.	[155]
Pramipexole	In vivo	Administered in 6-OHDA rats	Increases *Nurr1* expression and impedes the elevated expression of NF-κB and α-synuclein.	[198]
NURR1/RXR agonist HX600	In vitro	Reactive microglia, then exposed to LPS	Reduce the expression of nitric oxide synthase 2, macrophage receptor with collagenous structure, IL-1β, IL-6, and matrix metalloproteinase-9 and prevent inflammation-induced neuronal death	[160]
RXR agonist (IRX4204)	In vivo	Administered in 6-OHDA rats	Promote the survival of dopaminergic neurons in the SNc by activating cellular RXR-*Nurr1* signaling	[199]
Retinoic acid-loaded polymeric nanoparticles	In vivo	Administered in MPTP mouse	Increase the expression levels of *Nurr1* and Pitx3	[200]
Bupleurum falcatum, Paeonia suffruticosa, and Angelica dahurica	In vivo and vitro	Administered in MPTP-induced mice and PC12 cells	Increase NURR1 expression in the SNc and protect the dopaminergic neurons.	[141]
Ginkgo biloba extract (EGb761)	In vivo	Administered in MPTP-induced mice	Regulate the expression of *Nurr1*	[201]
Radix astragali ingredients (astragalus polysaccharide astraisoflavan)	In vitro	Administration in rat NSCs	Promote the expressions of sonic hedgehog, *Nurr1*, and Pitx3 mRNAs, and the proliferation of NSCs and induce NSCs differentiation toward dopaminergic neurons	[202]
**Neuroprotective and anti-inflammatory role of NURR1 in genetic engineering and stem cell therapy form**				
**Genes**	**Models**	**Methods**	**Main Findings**	**Ref.**
*Nurr1*	In vitro	Overexpression in LPS-induced reactive microglia	Reduce the expression of pro-inflammatory cytokines (TNF-α, ILβ)	[189]
		Overexpression in rat NSCs	Promote dopaminergic neuronal differentiation	[203]
		Overexpression in mouse OBSCs	Generate mature-like mesencephalic dopaminergic neurons and a subpopulation of dopaminergic-gamma-aminobutyric acid neurons under long-term culture conditions.	[204]
*Nurr1 +* NRBE	In vitro	P19 embryonal carcinoma stem cells transfected by *Nurr1*, then exposed to NRBE	Induce P19 stem cell differentiation into dopaminergic neurons	[205]
*Nurr1 + Pitx3*	In vitro	Overexpression in mouse iPSCs	Program iPSCs into functional dopaminergic-like neurons.	[206]
		Combined transduction of *Nurr1* and *Pitx3* in murine and human embryonic stem cell cultures.	Synergistically promote terminal maturation to the midbrain dopaminergic neuron phenotype	[204,207]
NURR1 + BRN4	In vivo	Co-transfect NSCs with Nurr1 and Brn4, then transplant into 6-OHDA rats	Increase the viability and maturity of dopaminergic neurons as well as the DA levels	[208]
*Nurr1 + Ngn2*	In vitro	Co-transduced mouse embryonic OBSCs	Reduce the proportion of TH-positive neurons	[204]
		Overexpression in the mouse midbrain progenitors	Increased production of dopaminergic neurons from midbrain progenitor cells	[209]
		Co-expression in rat and mouse NSCs	Repress *Nurr1*-induced generation of TH+ cells in rat cultures, enhance *Nurr1*-induced dopaminergic cell yields in mouse NPCs	[210]
*Nurr1 +* Ascl1	In vitro	Co-transduction and treatment by BDNF and neurotrophin-3	Increased *Nurr1*-induced production of dopaminergic neurons	[203]
*Nurr1 + Foxa2*	In vitro	Co-transduction in mouse NPCs	Induce the generation and differentiation of dopaminergic neurons, and their survival and resistance to toxic insult	[211]
	In vivo	Co-transduction in 6-OHDA rats		

Abbreviations: 6-OHDA: 6-hydroxydopamine; AQ: amodiaquine; ASCL1: achaete-scute complex homolog 1; BDNF: brain-derived neurotrophic factor; Brn4: brain 4; CQ: chloroquine; DA: dopamine; FOXA2: forkhead box protein A2; GSK-3β: glycogen synthase kinase-3 beta; HCQ: hydroxychloroquine; IL-1b: interleukin 1 beta; IL-6: interleukin-6; iPSCs: induced pluripotent stem cells; LBD: ligand binding domain; LPS: lipopolysaccharide; MPTP: 1methyl-4-phenyl-1,2,3,6-tetrahydropyridine; NF-κB: nuclear factor-kappa B; Ngn2: neurogenin 2; NPCs: neural precursor cells; NRBE: neonatal rat brain extract; NSCs: neural stem cells; OBSCs: olfactory bulb stem cells; Pitx3: pituitary homeobox 3; RXR: retinoid X receptor; SNc: substantia nigra pars compacta; TNF-a: tumor necrosis factor-alpha.

## 5. Conclusions and Perspective

Chronic inflammation can contribute to the pathogenesis and progression of PD either directly or indirectly. A proposed scenario is a central and peripheral interaction that leads to persistent inflammation and neurodegeneration. The inflammatory process in PD may downregulate NURR1 expression, thereby increasing the deposition of molecules and α-synuclein, thus exacerbating the inflammatory process and forming a vicious cycle. Given the essential role of inflammation in PD pathogenesis, immune-modulating treatments have emerged as important research targets. Although suppressing neuroinflammation alone or in combination with other medications may not affect the underlying etiology of the disease, it may limit the generation of neurotoxic compounds and improve treatment outcomes. NURR1-activating compounds, NURR1-modulators, and NURR1-based gene therapies are potentially efficient in treating PD. These treatments have demonstrated their efficacy by raising the expression of DA-related genes, protecting or repairing dopaminergic neurons from neurotoxins, inhibiting microglia activation, and impeding neuroinflammation. Despite these intriguing studies, the pathological changes and molecular mechanisms that relate NURR1 to the disease’s inflammatory process are still not fully understood. A deeper elucidation of the inflammatory mechanisms underlying the earliest signs of PD may one day make it possible to compete successfully with novel anti-inflammatory and immunomodulatory therapeutic approaches to slow or postpone disease progression. Recognizing the essential involvement of NURR1 in the inflammatory alterations associated with PD is critical for establishing a tolerable and timing safety window for NURR1-targeted therapy. Discovering small-molecule compounds that can effectively activate NURR1 either by directly binding to specific NURR1 sites or by modifying NURR1-related signal transduction should be the primary focus of any subsequent research.

## Figures and Tables

**Figure 1 ijms-23-16184-f001:**
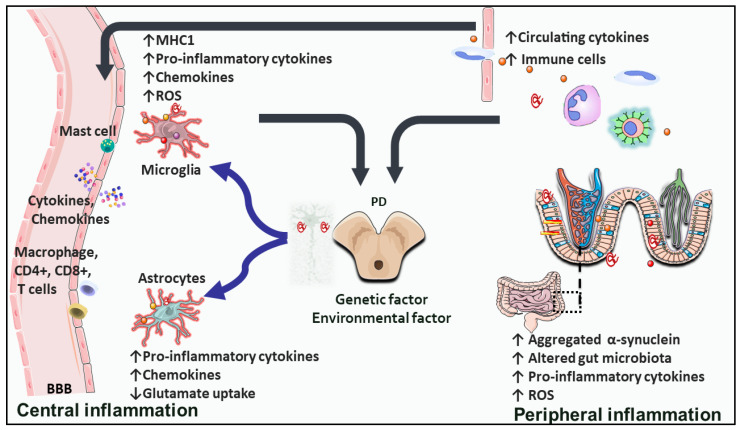
The figure highlights the inflammatory process in PD (↑: increase; ↓: decrease). The crosstalk between different cell types and intestinal dysbiosis with peripheral inflammation leads to increased circulating pro-inflammatory cytokines and immune cell activation. The hallmarks of a pro-inflammatory immune phenotype in PD include crossing the BBB and infiltrating immune cells from the periphery into the CNS. The two wavy, blue arrows indicate that dead or injured dopaminergic neurons and misfolded or aggregated proteins activate microglia and astrocytes, potentially resulting in a vicious circle. Abbreviations: BBB: blood-brain barrier; MHCI: major histocompatibility complex I; PD: Parkinson’s disease; ROS: reactive oxygen species.

**Figure 2 ijms-23-16184-f002:**
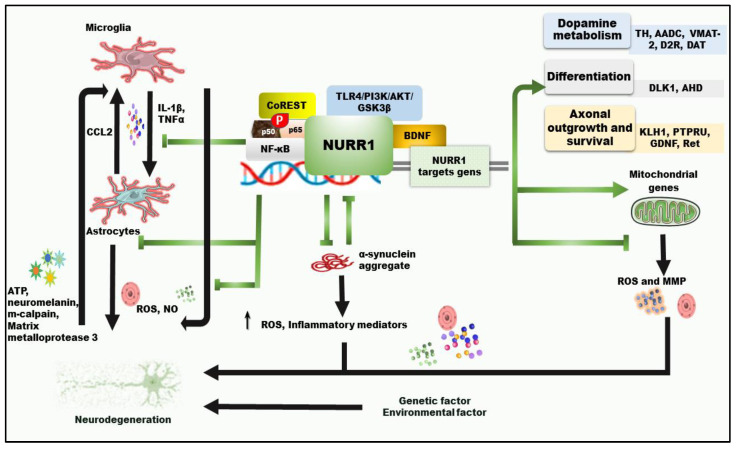
NURR1-regulated Neuroinflammation associated with PD. The green arrows indicate the regulatory mechanism of NURR1: “sharp head (positive regulation), blunt head (negative regulation).” The black arrows are the main inflammatory pathways; NURR1 is involved in PD. α-Synuclein and NURR1 have a bidirectional effect. α-Synuclein boosts inflammatory mediators and free radicals, enhancing α-synuclein aggregation, resulting in a vicious cycle. NURR1 positively regulates several nuclear-encoded mitochondrial genes and protects cells from mitochondrial membranes and ROS. Abbreviations: ATP: adenosine triphosphate; CCL2: chemokine ligand 2; MMP: mitochondrial membrane potential; NURR1: nuclear receptor-related transcription factor 1; ROS: reactive oxygen species.

**Table 1 ijms-23-16184-t001:** Central and peripheral inflammatory changes (cellular and molecular) in PD.

**PD Central Cellular and Molecular Inflammatory Changes**			
**Models/Sample**	**Methods**	**Main Outcomes**	**Ref.**
Human SNc	Compare PD to HC (Immunohistochemistry)	- Increase HLA-DR+ reactive microglia.	[26]
Human SNc	Compare PD to HC (Immunohistochemistry)	- Increase CR3/43 expression on reactive microglia in PD SNc.	[57]
Human striatum and SNc	Compare PD to HC (Immunohistochemistry)	- The number of CR3/43+ microglia is higher in PD patients. - Reactive microglia express TNF-α and IL-6.	[27]
Human striatum	Compare PD to HC (ELISA)	- Increase the levels of TNF-α, IL-1β, IL-2, IL-4, IL-6, epidermal growth factor, transforming growth factor-α, and transforming growth factor-β1 in the striatum of PD patients.	[58]
Human striatum and SNc	Compare PD to HC (Immunohistochemistry)	- Up-regulate nitric oxide synthase- and cyclooxygenase-1/2-containing amoeboid microglia of PD patients.	[59]
Human SNc	Compare PD to HC (Immunohistochemistry and Western blot)	- Increase recombinant C-X-C chemokine receptor type 4 and C-X-C chemokine ligand 12 of PD patients.	[60]
Human SNc	Compare PD to HC (Immunohistochemistry)	- Increase TNF-α glial cells in SNc of PD patients.	[61]
Human SNc	Compare PD to HC (Immunohistochemistry)	- Increase CD68+ amoeboid microglia in the SNc of PD patients.	[62]
Human SNc	Compare PD to HC (Immunohistochemistry)	- Increase the density of CD8+ and CD4+ T cells in the SNc of PD patients.	[63]
Mice striatum after α-synuclein seeds	Inject human PFF α-synuclein seeds or monomer	- Increase the immunoreactive area of Iba-1+ cells in the SNc. - Promote astrocytic proliferation.	[64]
Mice nigrostriatal after oral DSS	Oral administration of DSS	- Increased the nigral IL-1β and renin-angiotensin system pro-inflammatory activity. - Induce nigrostriatal dopaminergic neuron death. - Acute or sub-chronic gut inflammation.	[65]
Mice striatum after α-synuclein seeds injection	Inject α-synuclein into the striatum	- Increase expression of striatal IL-1α, TNF-α and IFN-γ.	[66]
Mice SNc after injecting LPS	Inject LPS into the 6-OHDA mice model	- Shift the microglia to phagocytic phenotype. - Reactive microglia with ED-1 and MHCII+. - Increase IL-1β secretion.	[67]
The mRNA profiles of microglia genes of mice	Compare microglia of specific pathogen-free and GF mice by flow cytometry and deep quantitative sequencing of the RNA transcripts	- GF mice display global defects in microglia with altered cell proportions and an immature phenotype. - Down-regulate the IL-1α. - Decrease the MHC I related β2 microglobulin in GF mice.	[68]
Mice SNc, midbrain after immune transfer cells	Adoptive transfer of copolymer-1 immune cells to MPTP mice model	- T cell accumulation in SNc. - Suppress the expression of macrophage antigen 1 in the midbrain. - Increase the expression of glial-derived neurotrophic factor.	[69]
Mice midbrain after transfer of Treg cells	Adoptive transfer of Treg cells to MPTP mice model	- Decrease the expression of TNF-α and inducible nitric oxide synthase mRNA in MPTP/Treg-injected mice.	[70]
Recombinant α-synuclein to heterozygous Tlr-4+/− mice	Homeostatic astrocytes were incubated with recombinant α-synuclein	- Reactivate astrocytes Tlr-4.	[71]
Rat striatum and SNc	Inject 6-OHDA in rat striatum	- Increase the uptake of 11C-PK11195 in the striatum and SNc.	[72]
Rat striatum	Inject 6-OHDA in rat striatum	- Up-regulate the genes of proteasome subunit beta type 8, MHC II, MIP-1α, colony-stimulating factor 2 receptor, and complement3/1q receptor.	[73]
Rat striatum	Inject 6-OHDA in rat striatum	- Increase striatal intercellular adhesion molecule 1 and platelet endothelial cell adhesion molecule-1 expression.	[74]
Monkey SNc	Intracarotid infusion of MPTP in rhesus monkeys	- HLA-DR+ reactive microglia in the SNc. - Extensive loss of dopaminergic neurons.	[75]
Monkey SNc	Inject MPTP in the vein of cynomolgus monkeys	- Increase HLA-DR+ microglia and astrocytes in SNc.	[76]
**PD Peripheral Cellular and Molecular Inflammatory Changes**			
**Models/Sample**	**Methods**	**Main Outcomes**	**Ref.**
Human PBMCs	Assess activation status by Flow cytometry in PD patients and HC	- Decrease the naive CD4+ and naive CD8+ T cells in PD patients. - Increase the memory of CD4+ T cells in PD patients. - Increase IL-17-producing CD4+ Th17 cells, IL-4-producing CD4+ Th2 cells, and IFN-γ-producing CD8+ T cells in PD patients.	[77]
Human whole blood	Single-cell transcriptome and T cell receptor sequencing of PD patients and HC.	- Increase CD8+T cells in PD patients.	[78]
Human serum	Assess serum cytokine concentrations in PD patients and HC by flow cytometry	- Increase cytokine levels IL-2, IL-4, IL6, IL-10, IFN-γ, and TNFα in PD patients.	[79]
Human peripheral blood	Analysis of the peripheral T-lymphocyte populations in PD patients and HC	- Decrease the ratios of CD4+:CD8+T cells. - Decrease CD4+ and CD25+ T cells. - Increase the ratios of IFN-γ-producing to IL-4-producing T cells.	[80]
Human PBMCs	Assess the cytokines expression in PD patients, HC, and non-PD neurological disease controls	- Increase TNF-α, IL-1β, IL-4, IL-6, and IL-10 in PBMCs in PD patients.	[81]
Human peripheral blood	Meta-analysis to assess the correlation between lymphocyte and natural killer cells in PD patients and HC.	- Decrease the numbers of CD3+ and CD4+ lymphocyte subsets. - Increase the number of natural killer cells in PD patients.	[82]
Human PBMCs	Assess cytokines by flow cytometry in PD patients and HC.	- Increase the expression of leucine-rich repeat kinase 2 in B cells, T cells, and CD16+ monocytes in PD patients.	[83]
Human fecal	Assess the intestinal inflammation fecal markers by ELISA	- Increase the fecal marker of intestinal inflammation (Calprotectin) in PD patients. - Increase intestinal permeability (alpha-1-antitrypsin and zonulin) in PD patients.	[84]
Human PBMCs	Investigate the levels of production and expression of cyto/chemokines by PBMCs in PD patients and HC through RT-PCR and ELISA	- Increase the levels of monocyte chemoattractant protein-1, regulated upon activation of normal T cell expressed and secreted, MIP-1α, IL-8, IFN-γ, IL-1β and TNFα in PBMCs in PD patients.	[85]
Leukocytes mice spleen after α-synuclein seeds in the striatum	Inject human PFF α-synuclein seeds or monomers into the striatum of mice	- Increase the total number of leukocytes within the spleen.	[64]
The colon of mice after injecting 6-OHDA	Inject 6-OHDA in the striatum and nigra of mice	- Increase IL-1β in the colon.	[65]
Mice DMV after administration of rotenone	Enteral administration of rotenone (gastric tube)	- Increase MHC II (M5/114.15.2) in the DMV of rotenone-treated mice.	[86]
Reactive microglia after being treated by MPP+ Mice SNc and serum after MPTP	MPP+ in reactive microglia LPS PD model MPTP was given subcutaneously in mice	- Increase NLRP3 expression, and the release of IL-1β and IL-18 in MPP+ treated microglia. - Increase the expression of Iba-1 level in MPTP mice SNc. - Increase the expression of IL-1β in MPTP mice serum	[87]
Mice midbrain after injecting LPS	Inject LPS in SNc of A53Ttg/tg mice and MPTP/p-induced PD mice	- Increase the expression of pro-cysteinyl aspartate proteinases-1 and IL-1β in midbrain homogenates of PD mice. - Increase the levels of NLRP3 in A53Ttg/tg mice midbrain.	[88]
Rat striatum and SNc after administration of rotenone	Rotenone chronically administers using subcutaneous osmotic minipumps	- Decrease the tyrosine hydroxylase immunoreactivity in the striatum and SNc.	[89]

Abbreviations: 6-OHDA: 6-hydroxydopamine; DMV: dorsal motor nucleus of the vagus; DSS: dextran sulfate sodium salt; ELISA: enzyme-linked immunosorbent assay; GF: germ free; HC: healthy controls; HLA-DR+: human leukocyte antigen D-related isotype; Iba-1: ionized calcium-binding adaptor molecule 1; IFN-γ: interferon-gamma; IL-1α: interleukin-1 alpha; IL-1β: interleukin-1 beta; IL-2: interleukin-2; IL-4: interleukin-4; IL-6: interleukin-6; IL-8: interleukin-8; IL-10: interleukin-10; IL-17: interleukin-17; IL-18: interleukin-18; LPS: lipopolysaccharide; MHC I/II: major histocompatibility complex class I/II; MIP-1α:macrophage inflammatory peptide-1α; MPP+: 1-methyl-4-phenylpyridinium; MPTP: 1-methyl-4-phenyl-1,2,3,6-tetrahydropyridine; NLRP3: NLR family, pyrin domain containing 3; PBMCs: peripheral blood mononuclear cells; PD: Parkinson’s disease; PFF: preformed fibrils; RT-PCR: real-time reverse transcription polymerase chain reaction; SNc: substantia nigra pars compacta; Th2: T helper 2; Th 17: T helper 17; Tlr4: toll-like receptor-4 pathway; TNF-α: tumor necrosis factor-alpha; Treg cells: regulatory T cells.

## Data Availability

Not applicable.

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
