# Peer review of "Advances in NURR1-Regulated Neuroinflammation Associated with Parkinson’s Disease"

_ijms, 2022, doi:10.3390/ijms232416184_

Round 1

Reviewer 1 Report

The present review is very interesting since it shows that the transcription factor NURR-1 is closely involved in one of the most relevant neurodegenerative diseases, Parkinson's disease. The data provided by the authors on the therapeutic potential for PD treatment of NURR1 modulating agents is also very interesting. However, the work contains important errors that must be corrected for publication.

In general, the sentences are very long, not clear and different concepts are mixed. It makes difficult to follow the idea of ​​what the authors intend to explain.

There are molecules that are cited without any explanation (for example, Kir6.1, page 3, line 112) or abbreviations that are used without prior explanation of them (for example, TFs, page 2, line 68). And like these examples, several by the text.

More specifically, there are misconceptions such as the claim that PD treatments are only symptomatic due to our incomplete picture of the exact pathophysionological and molecular basis of PD (page 2, lines 48-50). It is well known that, one of the reasons why the treatments are symptomatic is that when the disease is diagnosed, most of the dopaminergic neurons have degenerated, hence the importance of biomarkers for the early detection of PD.

Also noteworthy is the beginning of the abstract "Recent evidences.." which is quite incorrect since the relationship between neuroinflammation and neurodegenerative diseases has been known for a long time.

Also, authors should be updated with the nomenclature of microglia inflammation states, especially if a review on neuroinflammation is done.

Endothelial inflammation impact on PD section (page 4) should be rewritten as it is such a scant information for an important section.

Starting from section 3 (Nurr1), the information is quite dense since the authors cite many works, with many different molecules involved and, in most cases, they do not explain the function of that molecule or its relationship with the disease, including the full name. (Examples: EP1, BDNF, CREB, Ret protein, TLR9, HC...).

Some English or type errors: N line 349, they line 145, We and other studies line 179, neurodenerative disease line 193) and the lack of numbering in the section "Involvement of nuclear receptor-related factor 1 in PD neuroprotection and neuroinflammation" .

Author Response

ijms-2018610                                                                                Dec 2, 2022

Dear Professor Dr Cesar Borlongan and respected reviewers,

We greatly appreciate your handling and the constructive comments from the reviewers on our manuscript. We have revised the manuscript and checked it carefully according to the comments raised by the reviewers as listed below and highlighted with track changes in our revised manuscript. We hope that the current version is acceptable for your journal.

Sincerely yours,

Professor Dr Weidong Le

Professor and Director, Liaoning Provincial Key Laboratory for Research on the Pathogenic Mechanisms of Neurological Diseases, The First Affiliated Hospital, Dalian Medical University

Professor and Director, Institutes of Neurology, Sichuan Academy of Medical Sciences & Sichuan Provincial People’s Hospital, China Tel: +86-411-88135850 E-mail: wdle@sibs.ac.cn

………………………………………………………………………………………………….

Author's Notes to Reviewer

The present review is very interesting since it shows that the transcription factor NURR-1 is closely involved in one of the most relevant neurodegenerative diseases, Parkinson’s disease. The data provided by the authors on the therapeutic potential for PD treatment of NURR1 modulating agents is also very interesting.

Response: We thank the reviewer for the comment and evaluation.

Comment: However, the work contains important errors that must be corrected for publication. In general, the sentences are very long, not clear and different concepts are mixed. It makes difficult to follow the idea of ​​what the authors intend to explain.

Response: Thank you very much for your insightful suggestion to improve our review. Realizing that transcription factors impact and interact with many molecules. In the revised manuscript, we used your recommendation to focus on and streamline the critical thematic role of NURR1 and related molecules involved in developing PD inflammation. All changes were made by tracking changes.

Comment: There are molecules that are cited without any explanation (for example, Kir6.1, page 3, line 112) or abbreviations that are used without prior explanation of them (for example, TFs, page 2, line 68). And like these examples, several by the text.

Response: Thank you very much for your comment. We checked the manuscript to ensure that the molecules mentioned were explained. The abbreviations used throughout the manuscript have been double-checked to ensure there were defined in the first instance. The abbreviations are also listed in alphabetic order by the end of the manuscript. 

Comment: More specifically, there are misconceptions, such as the claim that PD treatments are only symptomatic due to our incomplete picture of the exact pathophysiological and molecular basis of PD (page 2, lines 48-50). It is well known that one of the reasons why the treatments are symptomatic is that when the disease is diagnosed, most of the dopaminergic neurons have degenerated, hence the importance of biomarkers for the early detection of PD.

Response: This is a much-appreciated correction, and we’ve revised it(page 2, lines 57-63).

Comment: Also noteworthy is the beginning of the abstract “Recent evidences..” which is quite incorrect since the relationship between neuroinflammation and neurodegenerative diseases has been known for a long time.

Response: This is a much-appreciated correction, and we have revised it.

Comment: Also, authors should be updated with the nomenclature of microglia inflammation states, especially if a review on neuroinflammation is done.

Response: Thank you very much for the update; In the revised version, we used the recently updated nomenclature. “Neuron. 2022 Nov 2;110(21):3458-3483. doi: 10.1016/j.neuron.2022.10.020. Microglia states and nomenclature: A field at its crossroads.

Comment: Endothelial inflammation impact on PD section (page 4) should be rewritten as it is such a scant information for an important section.

Response: Thank you very much for the comment; In the revised version, we updated and reinvented the crucial endothelial inflammatory discussion to be included.

Comment: Starting from section 3 (Nurr1), the information is quite dense since the authors cite many works, with many different molecules involved and, in most cases, they do not explain the function of that molecule or its relationship with the disease, including the full name. (Examples: EP1, BDNF, CREB, Ret protein, TLR9, HC...).

Response: We greatly appreciate your comment. We double-checked the NURR1 section to ensure that the molecules listed were explained and defined, as well as their functions and relationship to the disease. The abbreviations used throughout the manuscript have been double-checked to ensure they were defined the first time they were used. At the end of the manuscript, the abbreviations are also arranged alphabetically.

Comment: Some English or type errors: N line 349, they line 145, We and other studies line 179, neurodenerative disease line 193) and the lack of numbering in the section “Involvement of nuclear receptor-related factor 1 in PD neuroprotection and neuroinflammation”.

Response: Thank you very much for your comments. We revised the manuscript to ensure there were no typos, and a correction has been made to the numbering.

Reviewer 2 Report

The manuscript ijms-2018610 focuses on NURR1-regulated neuroinflammation in Parkinson's disease and can be interesting to specialists in this field. The Author's opinion is clear and based on good literature data. The review has a good structure, systematization of scientific data, and sequence of its presentation. The paper fits the Journal's scope.

However, I would suggest some revisions before publication :

-First of all, I would specify in the introduction that this is a narrative review, not a systematic review /meta-analysis.

-In line 231, the sub-heading (maybe 3.2 ?) is missing. Check also all the following sub-heading numbers.

-in line 4 probably an « and » is exceeding.

-Figures 1 and 2 descriptions should be improved. In figure 1, from reading only the caption, the meaning of the two small wavy green arrows pointing from PD to microglia and astrocytes is unclear; the reader needs to go back to the main text. The same for Figure 2. What are the differences between green and blue arrows? You also need to specify the symbols used for activation and inhibition.

-In the conclusion, I would further elaborate on the limitations of the current literature reviewed by the Authors, what needs to be done in the future and what are the take-home messages of the paper.

-Finally, I would further enrich the references by adding the following. : doi:10.3390/ijms23073476, doi:10.2174/1871527321666220310122415, doi:10.3233/JPD-223152

My decision is Minor Revisions

Author Response

ijms-2018610                                                                               Dec 2, 2022

Dear Professor Dr Cesar Borlongan and respected reviewers,

We greatly appreciate your handling and the constructive comments from the reviewers on our manuscript. We have revised the manuscript and checked it carefully according to the comments raised by the reviewers as listed below and highlighted with track changes in our revised manuscript. We hope that the current version is acceptable for your journal.

Sincerely yours,

Professor Dr Weidong Le

Professor and Director, Liaoning Provincial Key Laboratory for Research on the Pathogenic Mechanisms of Neurological Diseases, The First Affiliated Hospital, Dalian Medical University

Professor and Director, Institutes of Neurology, Sichuan Academy of Medical Sciences & Sichuan Provincial People’s Hospital, China Tel: +86-411-88135850 E-mail: wdle@sibs.ac.cn

………………………………………………………………………………………………….

Author's Notes to Reviewer

The manuscript ijms-2018610 focuses on NURR1-regulated neuroinflammation in Parkinson’s disease and can be interesting to specialists in this field. The Author’s opinion is clear and based on good literature data. The review has a good structure, systematization of scientific data, and sequence of its presentation. The paper fits the journal’s scope.

However, I would suggest some revisions before publication :

Comment: First of all, I would specify in the introduction that this is a narrative review, not a systematic review /meta-analysis.

Response: We thank the reviewer for this valuable correction. We clearly stated that the study was a narrative review.

Comment: In line 231, the sub-heading (maybe 3.2 ?) is missing. Check also all the following sub-heading numbers.

Response: We thank the reviewer for the correction. We corrected all the heading and sub-heading numbers.

Comment: In line 4, probably an « and » is exceeding.

Response: We thank the reviewer for the correction. We deleted.

Comment: Figures 1 and 2 descriptions should be improved. In figure 1, from reading only the caption, the meaning of the two small wavy green arrows pointing from PD to microglia and astrocytes is unclear; the reader needs to go back to the main text. The same for Figure 2. What are the differences between green and blue arrows? You also need to specify the symbols used for activation and inhibition.

Response: We appreciate the insightful comment; accordingly, figure 1 was revised to be clearer, and the caption was edited, including the description of the two wavy arrows. “The two wavy, blue arrows indicate that dead or injured dopaminergic neurons and misfolded or aggregated proteins activate microglia and astrocytes, potentially resulting in a vicious circle.” Figure 2 and the caption were revised, and the arrows’ color meanings were clarified. The green arrows indicate the regulatory mechanism of NURR1: “sharp head (positive regulation), blunt head (negative regulation).” The black arrows are the main inflammatory pathways; NURR1 is involved in PD.

Comment: In the conclusion, I would further elaborate on the limitations of the current literature reviewed by the Authors, what needs to be done in the future and what are the take-home messages of the paper.

Response: Thank you very much for the comment; we revised the conclusion and included the limitation and future perspectives. --------“Despite these intriguing studies, the pathological changes and molecular mechanisms that relate NURR1 to the disease’s inflammatory process are still not fully understood. A deeper elucidation of the inflammatory mechanisms underlying the earliest signs of PD may one day make it possible to compete successfully with novel anti-inflammatory and immunomodulatory therapeutic approaches to slow or postpone disease progression. Further research should concentrate on discovering small compounds that can efficiently activate NURR1 functions through direct binding to specific NURR1 sites or modulation of NURR1-related signaling.

Comment: Finally, I would further enrich the references by adding the following. : doi:10.3390/ijms23073476, doi:10.2174/1871527321666220310122415, doi:10.3233/JPD-223152

Response: We thank the reviewer for the update. We have updated the revised manuscript with the suggested articles doi:10.3390/ijms23073476, doi:10.2174/1871527321666220310122415, doi:10.3233/JPD-223152 as shown in references; 22, 183 and 52.

My decision is Minor Revisions

Reviewer 3 Report

The authors conducted a review on the role of neuroinflammation regulated by the NURR1 gene in Parkinson’s disease. The topic is highly relevant and I found the review to be clear and well written. The rationale underlying the choice to focus on NURR1 is well explained in the introduction. I only have a few comments to improve some parts.

-          The authors should specify that this is a narrative review

-          While the review is not a systematic review, I still think that adding one or two sentences on the criteria used by the authors to select the studies they are going to discuss would be helpful for readers

-          The Figure at page 6 should be named Figure 2

-          I liked how the two figures and the table summarized some of the key concepts presented in the review. I think that adding additional tables might help to better convey the content presented in the first sections of the review. 

-          Overall, I think it would be important to provide a more critical evaluation of studies (e.g. highlight whether the discussed studies have limitations in terms of design, sample size and so on) as at present it is not easy to understand which studies stand out based on quality of evidence according to the authors.

Author Response

ijms-2018610                                                                               Dec 2, 2022

Dear Professor Dr Cesar Borlongan and respected reviewers,

We greatly appreciate your handling and the constructive comments from the reviewers on our manuscript. We have revised the manuscript and checked it carefully according to the comments raised by the reviewers as listed below and highlighted with track changes in our revised manuscript. We hope that the current version is acceptable for your journal.

Sincerely yours,

Professor Dr Weidong Le

Professor and Director, Liaoning Provincial Key Laboratory for Research on the Pathogenic Mechanisms of Neurological Diseases, The First Affiliated Hospital, Dalian Medical University

Professor and Director, Institutes of Neurology, Sichuan Academy of Medical Sciences & Sichuan Provincial People’s Hospital, China Tel: +86-411-88135850 E-mail: wdle@sibs.ac.cn

.......................................................................................................................................................

Author's Notes to Reviewer

The authors reviewed the role of neuroinflammation regulated by the NURR1 gene in PD. The topic is highly relevant, and I found the review clear and well-written. The rationale underlying the choice to focus on NURR1 is well explained in the introduction. I only have a few comments to improve some parts.

Comment: The authors should specify that this is a narrative review

Response: We thank the reviewer for this valuable correction. In the revised manuscript, we clearly stated in both the abstract and introduction that the study was a narrative review.

Comment: While the review is not a systematic review, I still think that adding one or two sentences on the criteria used by the authors to select the studies they are going to discuss would be helpful for readers

Response: Thank you very much for your comment; we have added the criteria of the search strategy conduction of the review “A comprehensive PubMed search was conducted to find studies that assessed the role of inflammation in PD and NURR1 involvement that was published in English using the advance search (((Parkinson's disease) OR (PD)) AND ((Nurr1) OR (NR2A2))) AND (Neuroinflammation) / ((Parkinson disease) OR (PD)) AND (Neuroinflammation), by reading the title and abstract after that the closely related articles were full text read.”

Comment: The Figure at page 6 should be named Figure 2

Response: We thank the reviewer for this correction and correct it accordingly.

Comment: I liked how the two figures and the table summarized some of the key concepts presented in the review. I think that adding additional tables might help to better convey the content presented in the first sections of the review.

Response: Thank you for the comment; an additional table was added (Table 1) in the revised manuscript to summarize the crosstalk of central and peripheral inflammation in PD.

Comment: Overall, I think it would be important to provide a more critical evaluation of studies (e.g., highlight whether the discussed studies have limitations in terms of design, sample size and so on) as at present it is not easy to understand which studies stand out based on quality of evidence according to the authors.

Response: Thank you for the comment; in the revised manuscript, we have added a critical evaluation of the studies, including the limitation and future perspectives. All changes were made by tracking changes.

Round 2

Reviewer 1 Report

The present review has improved considerably after the extensive authors' corrections.